# A Gamified Digital Mental Health Intervention Across Six Sub-Saharan African Countries: A Cross-Sectional Evaluation of a Large-Scale Implementation

**DOI:** 10.3390/ijerph22081281

**Published:** 2025-08-15

**Authors:** Christopher K. Barkley, Charmaine N. Nyakonda, Kondwani Kuthyola, Polite Ndlovu, Devyn Lee, Andrew Dallos, Danny Kofi-Armah, Priscilla Obeng, Katherine G. Merrill

**Affiliations:** 1Grassroot Soccer, Hanover, NH 03755, USA; cnyakonda@grassrootsoccer.org (C.N.N.); kkuthyola@grassrootsoccer.org (K.K.); pndlovu@grassrootsoccer.org (P.N.); dlee@grassrootsoccer.org (D.L.); adallos@grassrootsoccer.org (A.D.); 2Viamo, Villa 21, Five Star Paradise, Runda, Nairobi 00619, Kenya; danny.kofi-armah@viamo.io (D.K.-A.); priscilla.obeng@viamo.io (P.O.); 3Center for Dissemination and Implementation Science, University of Illinois Chicago, 818 S. Wolcott Avenue, Chicago, IL 60612, USA; kgm@uic.edu

**Keywords:** mental health literacy, sub-Saharan Africa, adolescents, digital health, interactive voice response (IVR), gamification, mobile health (mHealth), low-resource settings, implementation science, mental health promotion

## Abstract

Mental health conditions affect many young people in sub-Saharan Africa (SSA), where stigma is high and access to care is limited. Digital tools accessible on basic mobile phones offer a scalable way to promote mental health, but evidence on their effectiveness in SSA is limited. This study evaluated the reach, feasibility, acceptability, and knowledge outcomes of Digital MindSKILLZ, an interactive voice response (IVR) mental health intervention implemented in the Democratic Republic of Congo, Ghana, Nigeria, Rwanda, Uganda, and Zambia. Over seven months, 700,138 people called the platform, and 425,395 (61%) listened to at least one message. Of these users, 63.6% were under 25 and 68.3% were from rural areas. The three content branches—mental health information, mental health skills, and soccer quizzes—were accessed by 36.5%, 46.4%, and 50.9% of users, respectively. Among users who accessed the mental health branch of the intervention, the mean number of messages completed was 7.6 out of 18 messages. In a follow-up survey, 91% of users understood the content, 85% would recommend the intervention, and 38% found the mental health content most helpful. Average knowledge scores were 62%, with lower scores on common disorders and stigma. The intervention showed strong reach and acceptability, but content and implementation improvements are needed to boost engagement and retention.

## 1. Introduction

Mental health difficulties disproportionately affect young people in Sub-Saharan Africa (SSA) compared to other global regions [1]. A recent systematic review found that adolescents (10–19 years) in SSA experience mental health conditions at higher rates than global and other low- and middle-income countries (LMICs) averages, with a prevalence of 26.9% for depression, 29.8% for anxiety disorders, 40.8% for emotional and behavioral problems, and 20.8% for suicidal ideation [2]. Globally, approximately 35% of mental health disorders onset by the age of 14 and 75% before the age of 25 years [3,4,5], making adolescence and young adulthood critical periods for the promotion of positive mental health and early intervention.

Increasing mental health awareness and enhancing mental health literacy are important strategies for promoting mental health and well-being among young people. Mental health literacy refers to “knowledge and beliefs about mental disorders that aid their recognition, management, or prevention” [6]. Greater mental health literacy is associated with more positive attitudes toward mental health professionals and increased help-seeking behaviors [7,8,9,10]. Conversely, limited mental health literacy has been linked to lower help-seeking and higher rates of anxiety and depression [11,12,13,14]. Adolescents and young adults have been identified as particularly important target groups for initiating and improving mental health awareness and literacy [15] to support young people in developing social and emotional foundations for mental well-being.

Despite its significance, mental health literacy among young people remains low worldwide [16], and SSA has some of the lowest levels of mental health awareness, especially among children and adolescents, who also report low confidence in mental health services [17]. There is a lack of population-level data on mental health awareness and limited mental health education campaigns for young people in SSA, contributing to ongoing stigma and limited help-seeking [17]. Low mental health literacy and stigma are compounded by sparse services and inadequate investment in mental health systems, particularly in rural areas [18,19]. Public health sectors in SSA lack sufficient funding, management, and services for mental health, with estimates that up to 90% of individuals with mental conditions in these regions do not receive any treatment [1].

Faced with a high mental health disease burden and a severe shortage of mental health workers, SSA has become a promising region for digital health innovation, increasingly turning to digital solutions for prevention and health system strengthening [20]. A meta-analysis of 144 digital mental health interventions (DMHIs) found positive effects on mental health outcomes, especially when digital tools were combined with in-person components [21]. These tools can help scale mental health support where in-person infrastructure is limited, especially in rural areas [22,23,24].

Digital health solutions designed with young people offer promising opportunities due to their wide reach, cost-effectiveness, and potential for anonymity [25], but the evidence base for DMHI in SSA remains limited. While millions of mental health applications exist globally, few are evidence-based, and even fewer have been evaluated in SSA [26]. Much of the current digital mental health evidence is from internet-based interventions often requiring smartphone access, which is a challenge in SSA, where smartphone penetration was only 55% in 2023, compared to 71% globally [27]. Among youth 15–24 in Africa, only an estimated 40% use the internet, and the percentage is lower in low-income countries in the region [28], with rural populations particularly disadvantaged due to barriers such as limited connectivity and the high cost of data [17].

To be effective, DMHI for young people must leverage the appropriate technology, be tailored to the local context, and respond to young people’s needs and interests to ensure acceptability and uptake [29]. One promising youth engagement strategy is gamification [30,31]. Gamification, or the use of game-based mechanics to improve engagement and learning [32], has been shown to increase engagement and motivation in digital health interventions [33]. When incorporating core mechanics such as feedback loops, narratives, and meaningful choices, gamification can support motivation and behavior change, particularly when strengthening users’ sense of autonomy and relatedness [34]. Localized and gamified interventions are not only important for uptake but can also enhance learning and retention.

There is a clear need and opportunity to develop and evaluate DMHI for young people, particularly in rural areas, that are accessible on basic mobile phones, provide relevant information using engaging strategies, and reflect the realities of connectivity, affordability, and technology access. Recognizing this opportunity for reaching young people in SSA, Grassroot Soccer (GRS) and Viamo partnered to develop Digital MindSKILLZ, an interactive voice response (IVR) mental health intervention accessible through Viamo’s platform. GRS is an adolescent health organization that uses a positive youth development approach, play-based education, and near-peer mentors to improve health and promote well-being. Since its founding in 2002, GRS has reached over 20 million young people with its evidence-based SKILLZ programs that use football (soccer) language and activities to convey critical health messages. GRS has demonstrated program effects through numerous studies on such topics as HIV prevention, gender-based violence, and substance use [35,36,37]. Viamo is a global social enterprise founded in 2012, specializing in leveraging mobile technology to deliver impactful information and services to underserved populations. Through its platform, Viamo has reached over 25 million people across 25 countries, utilizing voice-based technologies like IVR and SMS. By partnering with local telecom providers, Viamo ensures that its services are accessible on basic mobile phones, making it possible to reach remote and digitally disconnected communities.

The current paper presents an evaluation of the initial seven-month phase of the Digital MindSKILLZ intervention. The goal of the evaluation was to inform the future delivery of the intervention by addressing two aims:Aim 1: Assess the reach, feasibility, and acceptability of the intervention (i.e., the digital game) by country and user demographics;Aim 2: Assess knowledge of mental health topics among users based on responses during users’ engagement with the intervention.

Given the novelty of delivering a gamified mental health intervention via IVR in these settings, this evaluation was exploratory. We anticipated that pre-tested, localized, and gamified content might support user engagement and that knowledge gaps would emerge around basic mental health concepts and stigma based on several reviews of perceptions and mental health literacy of young people in LMIC and Africa [17,38].

## 2. Methods

### 2.1. Digital Intervention Overview

Digital MindSKILLZ is an IVR-adapted version of GRS’s MindSKILLZ program, an in-person mental health promotion intervention for adolescents. Viamo’s IVR platform is an automated telephone system that allows users to interact with pre-recorded audio content using their mobile phone keypad. When a user dials in, they are greeted with a menu of options and navigate the experience by pressing keys corresponding to different choices. Each selection triggers a specific voice recording, creating a branching, choose-your-own-adventure experience. Digital MindSKILLZ includes three main branches that users select from: mental health information, mental health skills (interactive coping strategies like guided breathing), and soccer-themed quizzes. The intervention includes 45 min of voice-recorded messaging across the whole intervention, and the entire experience is accessible on basic mobile phones, making it ideal for low-resource settings. An overview of the intervention’s storyboard is provided (Figure 1), along with sample audio clips from the intervention (see Appendix A.

Figure 1 is a visual representation of the main menu and overview of the user journey and intervention structure, including branching decision points and interactive elements. The text is from the intervention in Nigeria before translation into local languages.

In the mental health information branch, users can select one of four topic areas: “Me and My Mental Health,” “Keeping Good Mental Health,” “Know the Facts,” or “Getting Support.” Each topic leads to a series of short, voice-recorded messages covering key concepts such as the definition of mental health, identifying stressors, debunking myths, and strategies for seeking help. The full branch takes approximately 20 min to complete, with users able to return to the main menu, continue to the next branch, or exit the game after completing each topic. In the mental health skills branch, users choose from five mental health skills focused on relaxation techniques, emotional regulation, and strengths identification. Users learn the skill through guided instructions (1.5 to 3 min per skill, totaling 10–15 min). The soccer quiz branch of the intervention offers options for answering local and international soccer questions, with 5–6 questions in each section (totaling 5–10 min).

The intervention is gamified in a few ways. First, it embeds true/false quiz items within the mental health information branch, using a soccer announcer-style voice over, shouting “Goal!” or “Nice try!”, paired with crowd cheers or sighs to deliver immediate feedback. The intervention also incorporates soccer-themed examples when teaching relaxation and breathing techniques in the mental health skills branch. Finally, the soccer quiz branch is meant as a potential hook to draw in users who might not initially be attracted to mental health content but could engage after interacting with soccer-related material.

### 2.2. Intervention Development

Digital MindSKILLZ was developed using an iterative, youth-centered framework for designing digital health interventions [39], including the following steps:Step 1: Logic Model, Storyboard, and Scripts. The foundation of the Digital MindSKILLZ intervention was established through a structured logic model and scripted content. The logic model defined the intervention’s intended outcomes, including improved mental health knowledge, awareness of coping skills, and reduced stigma around mental health. The storyboard provided a visual representation of the user journey, mapping the flow of the intervention and key decision points. Finally, GRS drafted scripts that a group of “Coaches” (near-peer lay mental health providers) reviewed for cultural relevance and age-appropriateness.Step 2: Preliminary Audio-Recording and Youth Engagement. Once the scripts were finalized, preliminary voice recordings were produced using youth voice actors. A pilot evaluation of the draft intervention was conducted with young people in Lagos, Nigeria [40]. Participants (*n* = 25) played the intervention and then participated in focus group discussions to evaluate the content’s clarity, relevance, and impact. The pilot results informed key revisions, particularly in ensuring that mental health concepts were clearly defined and relatable, and including a broader range of mental health topics.Step 3: Refinement and Re-Recording. Based on user feedback, the intervention underwent content refinement. Additional mental health information and coping strategies were incorporated, including explanations of the differences between mental health and mental illness. The soccer quiz was expanded to include internationally relevant questions. Following these updates, the scripts were re-recorded to enhance clarity and engagement.Step 4: Contextualization and Translation. The Digital MindSKILLZ intervention was adapted for each country where it was implemented, including local Viamo teams that reviewed the content and facilitated translations into local languages. To further improve appropriateness, youth voice actors from each country recorded the localized versions.

### 2.3. Platform Promotion and User Onboarding

Digital MindSKILLZ is deployed through Viamo’s IVR platform, known widely as the 3-2-1 Service. The platform leverages partnerships with Mobile Network Operators (MNOs) to provide toll-free or low-cost access to users. To promote the platform, MNOs select target lists of users based on subscriber segments that match the platform’s value proposition, typically rural, lower-literacy, feature-phone users underserved by other channels. As a result, the user base is self-selecting, comprising individuals motivated enough to engage with voice-based information services.

Although Viamo’s platform is built on a shared technology backbone, its implementation is decentralized. Each country team works in close partnership with local MNOs and content partners to tailor platform configuration, promotion strategies, and content presentation. This decentralized approach means that each implementation varies in key factors such as airtime incentives, shortcode access, network quality, and user cost, which influence user recruitment, call success, and sustained engagement. Local partnerships promote the platform through SMS, voice blasts, and USSD push messages. In addition to digital outreach, content partners such as UNICEF or local government agencies promote the platform. These partners support awareness campaigns through radio spots, community events, and posters. In some instances, grassroots users themselves facilitate onboarding within their communities by hosting listening parties and encouraging referrals, further extending the platform’s organic reach.

First-time users are welcomed in their preferred language, introduced to the IVR system, and guided through an onboarding process. They are then invited to explore a menu of available content, including educational programs (such as Digital MindSKILLZ), entertainment, news, and games. After initial exposure to the platform’s offerings, users are prompted to opt in and optionally register by sharing demographic data (e.g., Do you agree to be contacted occasionally to receive information and to participate in surveys to share your opinion on matters important to you?). Returning users follow an adaptive journey tailored to their preferences and interaction history, receiving personalized content such as daily news, weather updates, or direct access to their most-used programs. In this case, Viamo users opt into the Digital MindSKILLZ intervention and can return directly to it in future sessions.

The intervention was rolled out in phases by country. Table 1 provides information about the launch date and number of days the intervention was “live” in each country during the evaluation period, along with the MNO and the intervention’s available languages.

### 2.4. Evaluation Design

This evaluation employed a descriptive, multi-country, cross-sectional design. The evaluation was conducted during the first seven months following the phased rollout across six countries: the Democratic Republic of Congo, Ghana, Nigeria, Rwanda, Uganda, and Zambia. We drew on the Implementation Outcomes Framework [41] to look at three implementation outcomes (Aim 1), which we defined as follows:Reach: The number, proportion, and characteristics of individuals who access Digital MindSKILLZ. We substituted the IOF’s outcome of penetration [41] with Glasgow’s definition for reach [42], given its greater applicability for delivery in community versus service settings.Feasibility: The extent to which Digital MindSKILLZ can be successfully implemented across countries in sub-Saharan Africa.Acceptability: The perception that Digital MindSKILLZ is agreeable, palatable, or satisfactory.

We also examined mental health knowledge based on users’ responses to true–false mental health statements embedded within the mental health information branch of the intervention (Aim 2).

### 2.5. Measures

Demographics: We measured participants’ age using the following age categories: under 18, 18–24, 25–34, 35–44, and 45 years and above. We measured gender (female or male), language, and geographic location using county and sub-county information as captured by mobile metadata or self-report. Geographic data, collected as Counties or Districts, were categorized as either urban or rural based on publicly available information.Reach: We measured the number of callers who dialed the platform and reached the main menu and the number of unique users—i.e., the number of individuals who accessed at least one voice recording during the intervention.Feasibility: We examined participant engagement as the average number of voice recordings accessed by each unique user. We also measured the number of unique users per full calendar month, disaggregated by country. We measured adherence as the number and proportion of messages completed in each of the three branches of the intervention (i.e., 18 messages in the information branch, 5 in the skills branch, and 11 in the soccer branch).Acceptability: We included three measures of acceptability: (1) comprehension: post-intervention user-reported understanding of the content, with three response options: “Understood all messages”, “Understood some messages”, and “Did not understand the messages”; (2) helpfulness: post-intervention user feedback on the most helpful aspect of the intervention, with the three response options: “Mental health information”, “Mental health skills”, and “Soccer quiz”; and (3) overall satisfaction: post-intervention rating of likelihood to recommend the intervention to a friend, with three response options: “Very likely”, “Not sure”, and “Very unlikely.”Mental health knowledge: We measured mental health knowledge based on users’ responses to 18 true-or-false statements embedded within the mental health information branch of the intervention. The statements were developed by GRS to align with the intervention’s content and objectives.

### 2.6. Data Collection Procedures

The evaluation dataset included all users who accessed the intervention between 1 September 2024 and 20 March 2025 and engaged with at least one voice-recorded message. The intervention platform enabled the real-time capture of user interaction and demographic data. Once a user accessed the platform, the system automatically captured key behavioral and technical data points—including the user’s phone number, unique contact ID, date and time of call, menu selections, content accessed, listening duration, and the user’s journey through the system. All passively collected data were initially stored locally on Delivery Nodes (DNs), which were securely hosted within the data centers of MNOs in each country. Periodically, data were synchronized with Viamo’s centralized Cloud Node (CN), where the data were aggregated and reviewed to inform platform enhancements, monitor engagement patterns, and support partner reporting needs.

In addition to passively collected data, users were given the option to voluntarily submit their age group, gender, and responses to three post-intervention questions. These prompts appeared during the intervention and clearly stated that providing demographic information was optional. All data were anonymous and linked only to a system-generated unique ID, never to a name or personally identifying information.

### 2.7. Ethical Considerations

This evaluation was determined not to involve human subjects research by the Office for the Protection of Research Subjects at the University of Illinois Chicago (Submission ID: STUDY2025-0614; determination issued 12 June 2025). All users interacted with the digital intervention voluntarily. No identifying information was collected. Users were not recruited for the evaluation, and there was no direct interaction between study personnel and users. All data collection adhered to Viamo’s user consent procedures and data privacy protocols.

### 2.8. Analyses

De-identified data were exported from Viamo’s centralized Cloud Node (CN) dashboard in Excel and CSV formats and shared with GRS for analysis. Descriptive and comparative analyses were conducted using SPSS (Version 30). Metrics were disaggregated by country, gender, and age group (i.e., under 18 and 18–24 years) to examine patterns and variations in user reach and engagement. Separate multivariate analyses of variance were conducted for each intervention branch, with age group, gender, and country as fixed factors and total messages listened to as the dependent variable. For any factor that showed a significant effect, pairwise comparisons were performed using the Games–Howell method to adjust for the unequal group sizes.

For missing demographic data, two approaches were used. For descriptive statistics (e.g., percent female), users who did not answer a given demographic question were excluded from both the numerator and denominator for that specific calculation. Overall, gender was reported by 65% of users and age group by 78%. For all inferential models predicting engagement (i.e., count of completed messages), listwise deletion was applied, where any user with missing data on any predictor (age or gender) was excluded, such that each model used only complete-case data. To assess the potential for selection bias, a sensitivity analysis was conducted comparing engagement levels between users with complete and incomplete demographic data. Although the difference was statistically significant (*p* < 0.001), the effect size was negligible (Cohen’s d = 0.08), suggesting that the analytic sample is broadly representative of the overall user base.

To examine mental health knowledge (Aim 2), a filter was used to identify users of the mental health information branch of the intervention. An individual-level dataset (*n* = 130,486) with unique IDs was generated and exported from Viamo’s cloud database. User responses to true–false statements at the end of each of the 18 messages were treated as baseline surveys, and descriptive statistics were used to summarize mental health knowledge data.

## 3. Results

### 3.1. Reach

A total of 700,138 people called into the platform, with 425,395 users (61.2% of the total callers) listening to at least one voice-recorded message. Among these users, 270,551 (63.6%) were youth under the age of 25 years. The flow and conversation rates from calling the platform to engaging with the specific components of the intervention are illustrated in Figure 2.

Figure 2 illustrates the conversion rates from calling the service to selecting among three content options: mental health information, mental health skills, and a soccer quiz. Users could interact with more than one branch during their call, so the percentages shown reflect overlapping participation and may sum to more than 100%.

Uganda accounted for nearly 60% of all users (Table 2). The gender distribution was generally balanced (46.4% female, 52.5% male), though country-specific differences were pronounced; for instance, only 22% of users in DRC were female compared to 66% in Rwanda. Despite being designed for youth, the intervention also reached adults aged 25 and older (*n* = 154,844, or 36.4% of the total users). The age distribution varied notably across countries: Rwanda had the youngest audience (80% under 25), while Uganda had the oldest, with 44% of users aged 25 or older. Notably, almost three-quarters of users were from rural areas, with percentages ranging from 62.6% in Uganda to 82.4% in Rwanda.

Table 2 presents disaggregated demographic data—gender, age, and rurality—of Digital MindSKILLZ users by country.

### 3.2. Feasibility

Participant engagement varied across the intervention’s three branches: 36.5% accessed mental health information, 46.4% engaged with guided mental health skills, and 50.9% participated in the soccer quiz (bottom of Figure 2). Among users, the average number of voice recordings completed was 5.4, equating to approximately 5–15 min of content. Country-level variation was substantial, with Nigeria demonstrating the highest average engagement and DRC the lowest (8.1 versus 3.7 recordings per user). Monthly trends revealed differing trajectories in users. In Uganda, the number of users fluctuated monthly but increased overall (Figure 3), whereas Nigeria and Rwanda saw peak engagement during months 3–4, followed by a steady decline, while the number of users in Zambia declined continuously after the launch (Figure 4), and Ghana had consistently low uptake. Although DRC was live for only two months at the time of data extraction, users nearly tripled from February to March.

Figure 3 displays the number of unique users accessing the intervention in Uganda per month during the evaluation period.

Figure 4 compares monthly trends in unique user engagement across Nigeria, Rwanda, and Zambia following the intervention’s launch in each country, and during the evaluation period.

### 3.3. Participant Engagement with Different Content Areas (Branches)

Among users who interacted with the mental health information branch, the average number of mental health messages completed was 7.6. In a multivariate ANOVA predicting levels of engagement from age, gender, and country, only country reached significance (F(5, 130,323) = 8.13, *p* < 0.001), with no effects of age (*p* = 0.54) or gender (*p* = 0.19). Games–Howell post hoc tests showed that Ugandan users completed more messages than DRC users (mean difference = +1.99 messages, *p* < 0.001) and more than Ghana (+1.31, *p* < 0.001). While differences were statistically significant due to large sample sizes, the differences in mean messages completed by country were modest. Attrition was steady (2–5% drop-off per message), leading to 16.7% (25,778) of users completing at least two-thirds of the branch and just 2.5% (3948) completing all 18 messages.

The soccer quiz, comprising 11 voice-recorded messages with true-or-false questions about local and international soccer trivia, averaged 4.4 of the 11 messages per user. Here, both country (F(5, 450,206) ≈ 520, *p* < 0.001) and age group (F(5, 450,206) ≈ 113, *p* < 0.001) were significant predictors of engagement, while gender remained non-significant (*p* = 0.12). Games–Howell comparisons for country ranked Rwanda highest; users completed 2.13 more quiz items than Zambians, which was ranked lowest (*p* < 0.001). For age, under-18 users engaged more than 45+ users by 0.54 items (*p* < 0.001) and 35–44 users by 0.56 (*p* < 0.001), with engagement declining progressively across older cohorts. Nearly 85% of users elected the local trivia option when given a choice.

The mental health skills branch was popular, with 46% of all users listening to at least one of the five voice-recorded messages. Most users were from Uganda (65%), followed by Nigeria (19%) and Rwanda (12%), with less engagement from users in Ghana, DRC, and Zambia. Among the five skills in the branch, users selected Power Hand most often (35%), the strength identification exercise, followed by the 3Ts (25%), Check-In (24%), Muscle Relaxation (9%), and Take 5 (7%). Age and gender trends mirrored overall engagement trends (e.g., 65% under 25, 54% male). Because the five skills were broken into multiple short audio clips, our “engagement per user” was too fragmented to treat as a single continuous outcome, preventing us from running multivariate tests.

### 3.4. Acceptability

A total of 134,439 of the 425,395 users (response rate = 31.6%) responded to post-intervention survey questions and reported high acceptability of the intervention. In response to a comprehension question, “Did you understand the game’s content?”, over 91% reported they understood some or all of the content. When asked, “What aspect of the game did you find most helpful?”, 38% selected mental health information, 33% the soccer-themed quiz, and 27% the mental health (coping) skills activities. When asked, “How likely are you to recommend the game to a friend?”, 85% of users reported that they were “very likely” to recommend it. While subgroup analyses by age and gender revealed minimal variation in intervention acceptability, country comparisons revealed that users from the DRC were noticeably less likely to recommend the game to a friend (57.6%) compared to users from other countries.

### 3.5. Mental Health Knowledge

On average, 55,000 users answered each of the 18 mental health knowledge questions in the information branch. Overall, 62% of responses were correct, indicating moderate awareness among users (Table 3). However, the proportion of correct responses varied considerably by item. Users demonstrated a high understanding of basic mental health practices and support-seeking behaviors, with correct responses reaching 93% for the item, “When a friend asks for help with a mental health problem, listening carefully is a great first step” (Item 16), and 89% for “Getting enough sleep is one of the best things we can do for good mental health” (Item 7). In contrast, misconceptions were observed around emotional expression, stigma, and the distinction between mental health and mental illness. For instance, 79% incorrectly equated good mental health with “Being happy all the time” (Item 1), while only 23% correctly disagreed with the statement, “Feeling sad means you have depression” (Item 14), and only 36% correctly rejected the idea that “Keeping feelings inside helps others understand what you need” (Item 4). Additionally, 66% of users agreed with the statement, “People with mental health problems are weak” (Item 11). When categorized by the thematic areas used within the intervention, users scored highest on “Keeping Good Mental Health” (mean = 76.2%) and “Getting Support” (65.0%), followed by “Know the Facts” (56.4%) and “Me and My Mental Health” (50.8%).

## 4. Discussion

Despite an increasing global interest in DMHIs, most evidence comes from high-income countries and internet-based interventions that require smartphones or consistent internet access [21,26]. This evaluation presents data collected over seven months on a large-scale, voice-based, gamified DMHI for young people in sub-Saharan Africa. We found very high reach, with over 700,000 people from six countries reaching the platform, and 425,000 people (270,551 under 25 years) listening to at least one voice-recorded message. The intervention’s reach among rural users, who comprised nearly three-quarters of all users, was particularly notable given recognized challenges of engaging rural youth in DMHIs [43].

Although the intervention’s self-directed options may have increased initial uptake and overall reach, attrition was high across the six countries, with 34% to 69% of users disengaging before listening to any mental health messages. Despite groups of highly engaged users, only 2.5% completed all 18 mental health messages, mirroring several digital mental health studies finding wide variability in user adherence and engagement [25,44,45], and challenges with attrition [45,46,47]. Several systematic reviews have documented consistently low adoption, poor adherence, and limited long-term engagement in DMHI deployed in real-world settings [44,48]. Yet, these reviews also highlight that a minority of “super-users” persists and that examining experiences among this group reveals critical enablers, such as personalized feedback loops, social support features, and culturally tailored content that boost sustained use. The reviews also identify recurring barriers, including technical malfunctions, privacy concerns, and low digital literacy, that impede DMHI [25,44]. Our use of the Implementation Outcomes Framework [41] allowed us to examine three key implementation outcomes that are useful to assess early on in projects and can predict adoption and sustainability of interventions [49]. However, our study would have been strengthened by using a determinants implementation science framework and mixed methods to explore the barriers and facilitators of implementation, shedding light on the how and why behind the implementation challenges we observed [50]. While we do not have qualitative insights from users, country differences in platform configuration, promotional strategies, and mobile network environments offer important clues about the observed variation in engagement. For example, in Nigeria, users completed an average of 8.1 messages, compared to just 3.7 in the DRC. Nigeria benefited from consistent promotion by the partnering MNO, broader network coverage, and higher call quality—all of which likely facilitated greater user retention. In contrast, DRC’s more fragmented infrastructure and limited promotion constrained access and engagement. Ghana, meanwhile, experienced repeated disruptions to its platform connection due to ownership and operational changes at its MNO (Vodafone Ghana, now Telecel), resulting in technical instability throughout the evaluation period. To address this, the Viamo platform will transition to MTN Ghana to provide more reliable connectivity and stable delivery.

Despite these challenges, Digital MindSKILLZ showed promise as a scalable DMHI, and the IVR platform may be a critical delivery channel for populations with low connectivity and limited utilization of web-enabled phones. On average, users completed 5.4 messages, and more for the users who accessed the mental health information branch of the intervention. The high reported acceptability is encouraging, as it is a strong predictor of future uptake [51]. The evaluation also revealed critical gaps in mental health knowledge among users at baseline. Some of the very low scores on basic mental health concepts (e.g., 21% for “good mental health means being happy all the time” or 23% “feeling sad means you have depression”) suggest possible fundamental misconceptions about mental health. Evidence from two separate systematic reviews shows that perceived lack of knowledge about mental health problems is one of the most prominent barriers to intended help-seeking among young people [13,52]. Interventions that highlight the positive attributes of mental health and assist young people in identifying mental health problems could improve both self-care and help-seeking.

In line with our goal of informing the future delivery of the Digital MindSKILLZ intervention, we generated the following recommendations for improving the intervention. We organized our recommendations according to the Implementation Outcomes Framework [41], while adding additional recommendations as they relate to the intervention’s effectiveness and future studies:

Recommendations for improving reach:Adapt Content and Delivery to Country-Specific Realities. Participant engagement varied by country, highlighting the need for tailored content and clarification about the factors contributing to these differences. Adapting the intervention’s content and implementation strategies for each setting, including language, voice actors, and examples, may enhance the intervention’s reach [53].Address the Digital Gender Gap. Overall, gender distribution among users was balanced (45.9% female), but gender disparities existed in some countries, most notably the DRC, where only 22% of users were female. These disparities align with regional trends, where females may face social and economic barriers to mobile ownership and access to digital tools [54,55]. However, Rwanda stands out as an example of how targeted strategies can promote more equitable access. On the Rwandan platform, 66% of users were female, a shift driven by deliberate efforts to prioritize women and girls, including gender-focused content, partnerships with women’s organizations, and tailored promotion strategies. Rwanda also benefits from higher mobile penetration among women and near-national platform coverage, which further facilitates equitable access. These results suggest that addressing the digital gender gap is not only feasible but essential for expanding reach. Future implementations should consider gender-inclusive approaches, including co-design with women’s groups, female-centric promotion, and ongoing monitoring of gendered engagement patterns.

Recommendations for improving feasibility:3.Optimize Onboarding to Improve Early Engagement and Adherence. Drop-off rates (i.e., call but do not listen to a voice recording) ranged from 34% to 69% across countries, underscoring the need for further investigation into user expectations, onboarding processes, and early-stage intervention design [25,49,56]. Enhancing the onboarding experience may boost the feasibility of the intervention. Conducting A/B testing is recommended to compare the current branching experience with a simplified linear model. Clarifying the intervention’s purpose and starting with engaging content has also been shown to retain users’ attention and reduce attrition, and should be incorporated into future onboarding strategies [48].4.Develop Models for Long-Term Engagement. Monthly users showed steady declines over time, with Uganda being a notable exception. Sustained engagement of DMHIs hinges on maintaining user engagement beyond initial exposure, and sustained engagement strategies, such as periodic promotion, content updates, or integration with other platforms or initiatives, should be explored.

Recommendations for improving the potential effectiveness of the intervention:5.Ensure Entertainment is a Bridge to Learning. User preferences suggest the soccer quiz was a key entry point, with more users engaging with it than with the mental health content. The soccer quiz and voice-activated feedback (e.g., “Goal!” or “Nice try!”) were intentionally designed as gamified elements to support user engagement. However, our findings highlight the need to ensure these mechanics function as bridges to health content rather than distractions. Future iterations should ensure that gamified components are meaningfully integrated with essential mental health messages.6.Use Validated Mental Health Measures Embedded in the Intervention. Correct responses to the 18 true/false items ranged widely, revealing possible knowledge gaps in critical mental health concepts among the intervention’s mostly young and rural users, a group typically underrepresented in mental health research [57,58]. However, future iterations must incorporate psychometrically validated measures (e.g., Mental Health Literacy Scale) to ensure accurate, reliable, and culturally appropriate measures of mental health outcomes.

Recommendations for future studies of the intervention:7.Plan Future Studies of Intervention Effectiveness and Implementation. Building on the results from this evaluation, a future evaluation of this intervention should use a mixed-methods and longitudinal approach to capture effectiveness and implementation outcomes. It should also draw on a determinants implementation science framework to explore barriers and facilitators to implementation, which was beyond the scope of this evaluation. Qualitative insights are required to explore many unanswered questions about user experiences and behaviors. Future evaluations should also include pre–post assessments with control or comparison groups and long-term follow-up. A pilot randomized controlled trial is tentatively planned in Malawi and will offer an opportunity to advance this work.

### Limitations

This evaluation had several significant limitations. First, the evaluation relied on a convenience sample of people who voluntarily engaged with the intervention, introducing self-selection bias and limiting the generalizability of the findings. More motivated, tech-savvy, or health-conscious young people may choose to engage and differ from the general youth population in important ways. Second, the cross-sectional evaluation design limits our ability to assess the intervention’s effectiveness in imparting mental health knowledge and skills, as it does not allow us to examine changes in user knowledge post-intervention. This limits our ability to determine the feasibility of this intervention as a public mental health intervention. Another important limitation is that the mental health knowledge items were developed to align with the intervention’s content rather than being drawn from psychometrically validated measures, limiting their validity and reliability in measuring mental health literacy. A third significant limitation was the lack of qualitative insights from users. Although the platform passively captured engagement and basic demographic data, it did not capture qualitative feedback. These insights could have deepened our understanding of the user experience with the intervention and barriers, and facilitators to access and engagement. Furthermore, the acceptability measures were not from a validated scale and were simplistic, providing limited insights.

## 5. Conclusions

The evaluation of Digital MindSKILLZ demonstrates the reach of an IVR-based intervention in delivering mental health information to large numbers of youth in low-resource settings across six sub-Saharan African countries. Despite high initial drop-off and variable adherence, over 425,000 users engaged with the intervention, most of them under 25 years and from rural areas. High acceptability and user feedback highlight the promise of the intervention for building mental health literacy. This evaluation’s insights provide a strong foundation for improving the intervention’s design and feasibility. Additionally, this evaluation will provide critical information that will inform the future study of the Digital MindSKILLZ intervention in Malawi, which is currently being planned.

## Figures and Tables

**Figure 1 ijerph-22-01281-f001:**
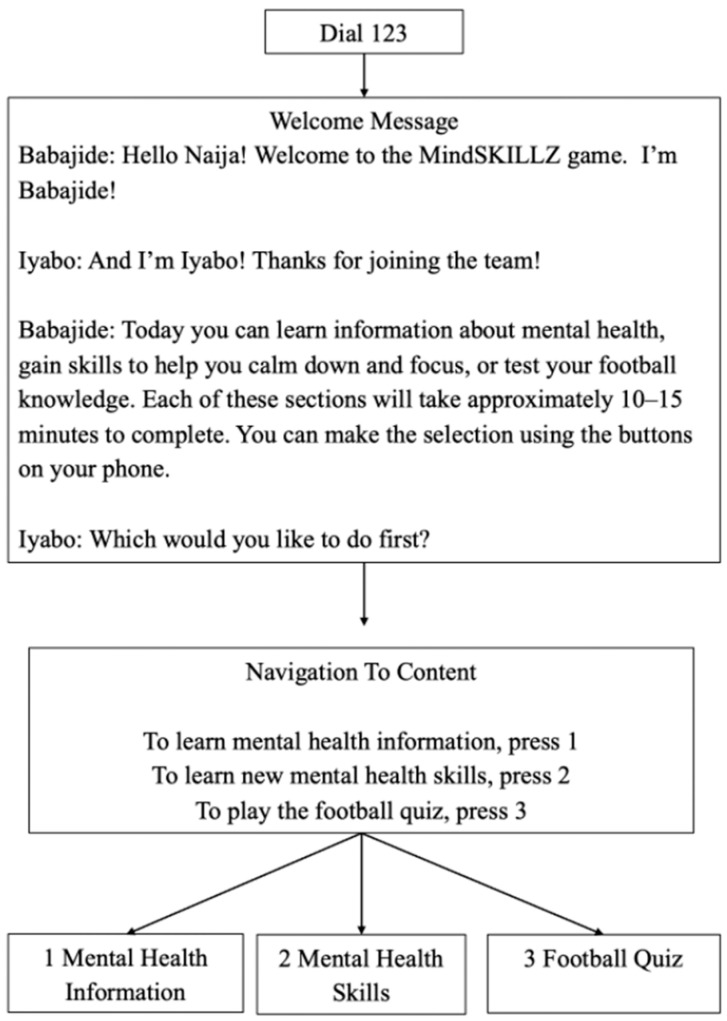
Digital MindSKILLZ: storyboard overview from Nigeria.

**Figure 2 ijerph-22-01281-f002:**
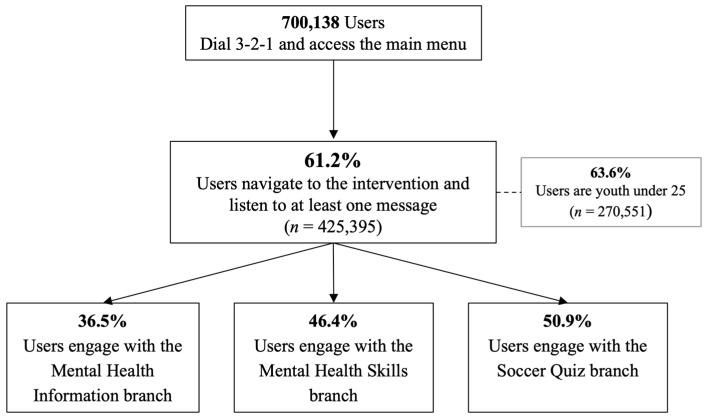
Conversion rates by the intervention’s content branches.

**Figure 3 ijerph-22-01281-f003:**
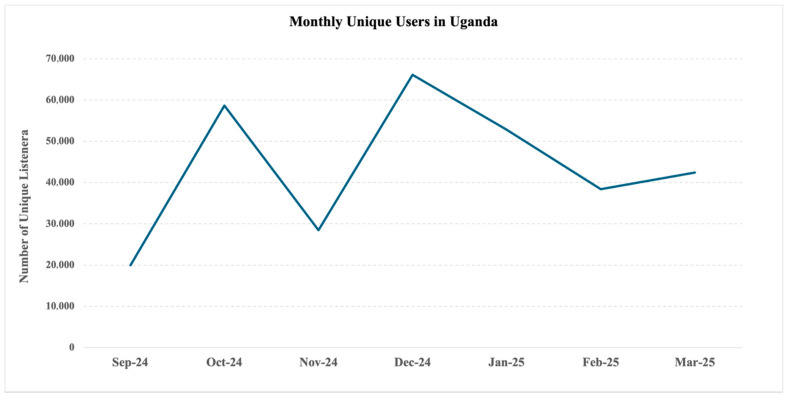
The number of monthly unique users in Uganda, September 2024 to March 2025.

**Figure 4 ijerph-22-01281-f004:**
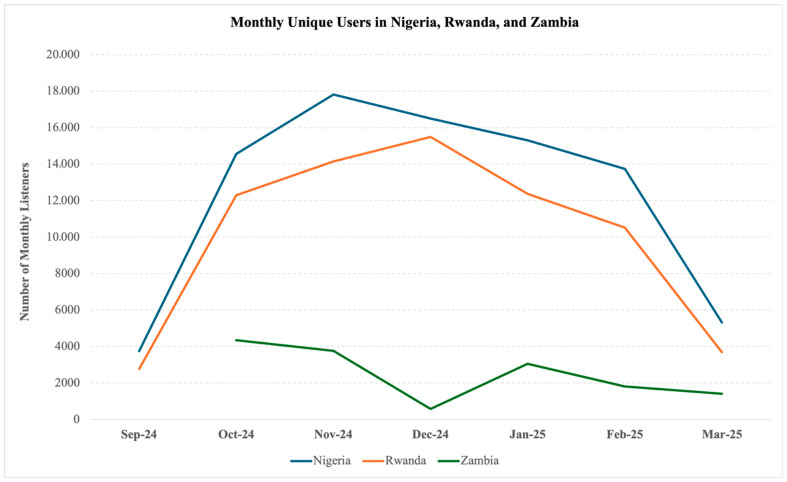
The number of monthly unique users in Nigeria, Rwanda, and Zambia, September 2024 to March 2025.

**Table 1 ijerph-22-01281-t001:** Digital MindSKILLZ rollout and operations.

Country	Mobile Network Operator (MNO)	Intervention Language Availability	Launch Date	Number of Days Active
DRC	Vodacom	Lingala	7 February 2025	46
Ghana	Vodafone	Twi	25 September 2024	176
Nigeria	Airtel	Pidgin, Hausa, Yoruba, Igbo	19 September 2024	182
Rwanda	MTN	Kinyarwanda	13 September 2024	188
Uganda	Airtel	Luganda	25 September 2024	176
Zambia	MTN	Bemba	7 October 2024	164

Table 1 details the launch date, duration of availability, MNO, and intervention language for each country where Digital MindSKILLZ was deployed. The number of days active was calculated as the number of days from the launch date to 20 March 2025, the period of data collection used for this evaluation.

**Table 2 ijerph-22-01281-t002:** User demographics by country.

Country	Unique Users	Unique Users (Percent of Total)	Female	Under 25	Rural
DRC	22,632	5.3%	22.4%	74.2%	76.8%
Ghana	950	0.2%	43.9%	71.7%	NA
Nigeria	76,282	17.9%	43.8%	68.4%	72.0%
Rwanda	63,654	15.0%	65.8%	80.4%	82.4%
Uganda	248,713	58.5%	41.7%	55.7%	62.6%
Zambia	13,164	3.1%	58.1%	70.2%	75.6%
Total/^a^ Mean	425,395	100%	46.4%	63.6%	68.3%

^a^ Weighted average.

**Table 3 ijerph-22-01281-t003:** Participant mean correct responses (%) by item and category.

#	Category and Items	* Percent Correct
Me and My Mental Health		50%
1	Having “good mental health” means being happy all the time.	21%
2	There are treatments for mental health conditions such as depression and anxiety.	75%
3	Everyone deserves support and understanding, even if their culture views mental health differently.	80%
4	Keeping your feelings inside helps others understand how you feel.	36%
5	When I experience strong emotions, they control me; there’s nothing I can do to calm down.	42%
Keeping Good Mental Health		76%
6	Stress is normal, but too much stress harms our minds and bodies.	83%
7	Getting enough sleep is one of the best things we can do for good mental health.	89%
8	Social media can also be a mental health risk.	82%
9	A healthy diet means eating meat at every meal.	35%
10	Setting goals and making plans to achieve them helps us grow into the people we want to be.	92%
Know The Facts		56%
11	People with mental health problems are weak.	34%
12	Mental health stigma can prevent people from seeking help.	66%
13	Negative peer pressure can make it hard to say no to drugs and alcohol.	74%
14	Feeling sad means you have depression.	23%
15	People experiencing anxiety may have different symptoms.	85%
Getting Support		65%
16	When a friend asks for help with a mental health problem, listening carefully is a great first step.	93%
17	There are mental health services available to you.	85%
18	You are a young person, so your privacy is not important.	17%
Overall Correct		62%

Table 3 shows the percentage of users who correctly answered each true-or-false mental health knowledge item, organized by thematic category from the intervention content. * Percentage of respondents providing the desired response.

## Data Availability

The data presented in this study are available on reasonable request from the corresponding author. The data are not publicly available due to privacy restrictions and data-sharing agreements with Grassroot Soccer, Viamo, and Mobile Network Operators.

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
