# Peer review of "A Gamified Digital Mental Health Intervention Across Six Sub-Saharan African Countries: A Cross-Sectional Evaluation of a Large-Scale Implementation"

_ijerph, 2025, doi:10.3390/ijerph22081281_

Round 1

Reviewer 1 Report

Comments and Suggestions for Authors

Dear Authors Christopher K Barkley , Charmaine N Nyakonda , Kondwani Kuthyola , Polite Ndlovu , Devyn Lee , Andrew Dallos , Danny Kofi-Armah , Priscilla Obeng , Katherine Merrill,

I would like to start off by saying that I think that this is an excellent and much needed study. As one of few large-scale, real world, digital mental health interventions in the Global South, and only the first of its kind across multiple countries in Sub-Saharan Africa, this work fills a critical gap in the field of global mental health and digital health implementation science. The sample size and scope of this intervention (over 700k total callers across six countries) is truly impressive and provides unique insights to feasibility of using basic cell phone infrastructure to deliver mental health support to people in low resource settings.

Summary

This cross-sectional manuscript uses the RE-AIM evaluation framework to evaluate reach, feasibility, acceptability and mental health knowledge outcomes among users of Digital MindSKILLZ, an IVR mental health game implemented across six Sub-Saharan African countries and received by 425,395 unique callers over seven months. Results demonstrate high reach of the intervention, particularly among rural youth (68.3% rural, 63.6% <25), and high acceptability (91% comprehension, 85% recommend to friend), but low adherence to and comprehension of the mental health content (7.6 of 18 messages completed, large gaps in mental health knowledge.

Strengths

Methodological Rigor and Scale

The scale of the sample and diversity of the implementation contexts (six countries across Sub-Saharan Africa) is an outstanding strength of the study. In addition, the use of passive data collection from the IVR platform allowed for a more real-time capture of data and user interactions without burden on participants (lines 261-271). Use of an evaluation framework grounded in implementation science outcomes (reach, feasibility, acceptability) is appropriate for this type of real-world intervention and use case.

Innovative Technology Application

The authors note the importance and appropriateness of their choice of technology for basic mobile phones, which can overcome a major technological barrier to using smartphone applications in Sub-Saharan Africa, where smartphone penetration is lower than global averages at 55% versus 71% respectively (line 80. This aspect of the intervention design demonstrates careful consideration of the target audience and contextual issues related to infrastructure and digital inequality.

Cultural Adaptation and Localization

I also like how the intervention itself was localized and carefully adapted using a youth-centered and iterative development process with local language translation and content localization (lines 176-181). The football-based engagement and game content also appears to be a culturally tailored element that should be highlighted and discussed further.

Areas for Improvement- Major Concerns

  1. Limited Outcome Measurement (Lines 254-257)

The use of embedded true/false questions as the only measure of mental health knowledge outcomes and the reliance on these outcomes to make inferences about intervention effects are major limitations of the study. The authors note that these items were not taken from any validated measures (lines 505-508), which severely undermines the construct validity and reliability of any claims based on these data. These 18 items had a large range in percentage of respondents who answered them correctly (17% to 93%), indicating possible issues with item construction, appropriateness or lack of validation (lines 517-533.

Recommendation: Future work should use validated mental health literacy measures (e.g., Mental Health Literacy Scale, MHLS) or, if creating new measures, ensure that they are psychometrically validated using proper validation procedures.

  1. Cross-Sectional Design Limitations (Lines 498-501)

The cross-sectional nature of the data and lack of ability to assess pre-post changes in knowledge or behavior is also a serious limitation of this study. This design can only assess patterns of engagement with the game and call conversion rates, as well as a baseline level of mental health knowledge among users, but it cannot demonstrate any potential impact of the intervention itself on these outcomes

Recommendation: The authors should consider a longitudinal study design with a matched comparison group, or at minimum pre-post assessments among a subsample of users to address this gap.

  1. High Attrition and Limited Engagement

Drop-off rates of 34-69% between the stage of calling the number and actually listening to the content (lines 438-441) and the low percentage of users who actually listened to all 18 mental health messages (2.5% completion rate) are both major limitations that the authors need to address. The question of why users stop engaging with the intervention and where there are large drop-offs (listening vs selecting branches) is an important question that the study design does not allow for.

Recommendation: Conduct qualitative research to identify reasons for non-completion and develop retention strategies.

Methodological Issues

  1. Demographic Data Quality (Lines 231-236)

The manuscript is not very clear on response rates for demographic questions (lines 236) and selection bias in who actually provided this information, or whether there were differences in response patterns between the included demographic questions. The dramatic country-level differences in gender balance among users (22% female in DRC vs 66% in Rwanda) also requires further explanation (lines 314-324)

Recommendation: Authors should provide more information about data quality and response rates for the demographic questions as well as sensitivity analyses to assess possible selection bias.

  1. Country-Level Variation Analysis

Country-specific data is presented, but there is insufficient analysis or discussion of the factors that may contribute to the large variation in engagement and completion rates between countries (3.7 messages completed in DRC vs 8.1 in Nigeria, lines 471-474)

Recommendation: Include analysis of country-level contextual factors (connectivity, promotion strategies, cultural factors, etc) that may explain these differences.

  1. Statistical Analysis Limitations

The statistical analysis, while largely descriptive, could have been expanded to consider bivariate and multivariate predictors of engagement and outcomes given the large sample size and number of variables

Recommendation: Use of multilevel modeling to account for clustering within countries and identify individual-level and contextual predictors of outcomes.

Minor Issues

  1. Reporting and Presentation

  • Line 24-25: The statement “average completion was 7.6 out of 18 messages” is potentially misleading as it seems to imply that this is the average across all users (suggesting low engagement) when the text indicates that this is specifically among those that selected the mental health branch. May need to reword for clarity.

  • Table 1, Line 210: The total days active calculation in this table should be checked for accuracy. Also, the end date in the March 20, 2025 looks inconsistent with the manuscript preparation date in June 2025.

  • Lines 376-384: In the acceptability section, it is not entirely clear how the response numbers are calculated based on the total number of users (134,439 respondents vs 425,395 total users, lines 378). It may be helpful to put this information in the flow diagram.

  • Figure 2: The flow diagram could include percentage calculations for each stage to better visualize the conversion rates from one stage to the next.

  1. Literature Integration

Literature on existing digital mental health interventions and implementation science frameworks should be better integrated throughout the discussion beyond just high level statistics

Specific Technical Comments

Lines 113-116: Aims of the evaluation are clearly stated but could benefit from being more explicit about any hypotheses or predictions about expected outcomes based on prior research or theory.

Lines 285-293: The analytical approach section could be expanded to more clearly describe how missing data was handled and any sensitivity analyses performed.

Lines 458-467: Interpretation of the mental health knowledge findings may be improved by providing more nuanced discussion of these results. Some of the very low scores on basic mental health concepts (e.g., 21% for “good mental health means being happy all the time”) suggest possible fundamental misconceptions about mental health that have implications for intervention design.

Recommendations for Revision

Major Revisions Needed:

  1. Strengthen outcome measurement: Acknowledge limitations of non-validated measures, and discuss implications for interpretation of findings.

  1. Enhance country-level analysis: Provide more detailed exploration of contextual factors contributing to country differences.

  1. Expand statistical analysis: Consider more sophisticated analytical approaches to identify predictors of engagement and outcomes.

  1. Improve reporting clarity: Address inconsistencies in data presentation and provide clearer flow of participant numbers.

Minor Revisions:

  1. Correct data presentation inconsistencies throughout tables and figures.

  1. Enhance discussion with deeper integration of relevant literature and theoretical frameworks.

  1. Clarify methods regarding data collection procedures and quality assurance measures.

Overall Assessment

This study makes an important contribution to the emerging field of digital mental health in Sub-Saharan Africa. The scale and reach of this intervention is very impressive and provides valuable proof-of-concept evidence for IVR-based interventions. However, the limitations of the study in terms of outcome measurement and study design prevent the authors from drawing stronger conclusions about intervention effectiveness. The work would be strengthened by addressing the above methodological concerns.

Despite these limitations, the study findings provide important and novel insights for future digital mental health interventions in low resource settings and contribute a needed dose of implementation science evidence to the field.

Rating: (comments below)

  • Novelty: High - first large scale IVR mental health intervention in SSA

  • Scope: Appropriate for journal

  • Significance: High - addresses critical gap in global mental health

  • Quality: Moderate - Good design, but methodological limitations

  • Scientific Soundness: Moderate - Appropriate methods, but outcome measurement concerns

  • Interest to Readers: High

  • Overall Merit: Moderate to High

  • English Level: Good

Recommendation: Major Revision

The manuscript has significant merit and is of interest to the journal, but would require substantial improvements in terms of outcome measurement discussion, analytical approach, and clarity of reporting to meet standards for publication in a high-quality journal.

Sincerely,

Serving peer reviewer at MDPI

Reviewer 2 Report

Comments and Suggestions for Authors

This manuscript, “A Gamified Digital Mental Health Intervention Across Six Sub-Saharan African Countries: A Cross-Sectional Evaluation of a Large-Scale Implementation”, reports on a timely public health initiative with several core strengths. Its novelty lies in evaluating a large-scale, gamified mental health intervention using Interactive Voice Response, an innovative approach to reach underserved youth and rural populations in Sub-Saharan Africa. The impressive scale, having reached over 700,138 callers, provides invaluable real-world data on the platform’s potential. Methodologically, the paper is well-structured, employing a youth-centered design framework and a rigorous implementation outcomes model to assess reach, feasibility, and acceptability. The results are presented with clarity, and the discussion is a particular highlight, offering specific, actionable recommendations for future improvements, which adds significantly to the paper’s practical value.

While the manuscript is strong, there are several concerns I have with this manuscript:

First, on a conceptual level, the application of “gamification” is underdeveloped. The paper identifies gamification as a key strategy but the intervention’s game-like elements appear to be limited to a football quiz. The manuscript would be strengthened by a more thorough engagement with gamification theory to explain how a quiz, in this context, functions as an effective engagement mechanic. The discussion insightfully questions whether the quiz served as a “bridge to learning” or a “distraction”, but this analysis would be more profound if it were grounded in established principles of gamification for health behavior change.

Second, the description of the ethical procedures requires significant clarification. The manuscript states that the evaluation was determined “not to involve human subjects research” by an IRB, yet it also reports that “Informed consent was obtained from all users who voluntarily provided demographic data”. These statements appear contradictory. The authors must clarify the distinction between the different types of data collected (passive analytics vs. actively solicited demographics) and the specific consent mechanism for each. The assertion that “no interaction with users occurred” is confusing in the context of an evaluation of an interactive digital tool and should be rephrased for clarity.

Third, the methodology relies on a convenience sample, which introduces a potential for self-selection bias that is not fully explored. Users were recruited through MNO promotions like SMS and voice blasts. It is unclear who receives these promotions and why they choose to engage. This limits the generalizability of the findings. The discussion should more explicitly address this limitation by considering how the self-selected user base might differ from the broader population of youth in these regions in terms of motivation, technological literacy, or other characteristics.

Fourth, there are issues with the reporting of results and the measures used. A paragraph describing user engagement with the mental health skills and football quiz branches is duplicated in the results section, an oversight that requires correction. Furthermore, while the very low uptake in Ghana is noted, it is a significant outlier that warrants a more detailed discussion of potential causes. The measures for acceptability – comprehension, helpfulness, and likelihood to recommend – are also quite simplistic. Forcing users to choose only one “most helpful” aspect, for example, provides limited insight. Acknowledging the limitations of these bespoke measures in comparison to validated scales would strengthen the paper.

Finally, the discussion of the intervention’s adherence and feasibility could be more critical and better integrated with implementation science theory. The authors rightly note the low completion rate for the full mental health curriculum (only 2.5% of users completed all 18 messages), but the implications of this finding are not fully explored. Can an intervention be considered feasible or have a public health impact with such low adherence to the core content? Moreover, while the recommendations in the discussion are excellent, they are not explicitly linked back to the implementation outcomes framework cited in the methods. Framing the challenges and recommendations using constructs from implementation science (e.g., relating the digital gender gap to factors in the “outer setting” or proposing A/B testing as a specific implementation strategy) would provide a more rigorous theoretical grounding for the discussion.

In sum, this manuscript details a valuable and large-scale evaluation of a promising digital health intervention. The study’s scale, novelty, and focus on vulnerable populations are major strengths. However, the manuscript requires major revisions before it can be considered for publication. The authors need to address the concerns outlined above, with particular attention to: deepening the conceptual discussion of gamification; clarifying the ethical procedures; addressing the limitations of the sampling strategy; correcting reporting errors and strengthening the measurement discussion; and providing a more critical, theoretically-grounded analysis of the intervention’s low adherence rates. Addressing these points will significantly strengthen the paper and its contribution to the field.

Round 2

Reviewer 1 Report

Comments and Suggestions for Authors

The current version of the manuscript is fundamentally improved. Thank you for considering my expertise. 

Author Response

Dear Reviewer,

Thank you again for your thoughtful review and valuable feedback throughout the process.

Warm regards,
Chris Barkley

Reviewer 2 Report

Comments and Suggestions for Authors

The authors have been exceptionally responsive to the feedback provided.  The manuscript is now a much stronger, more nuanced, and impactful piece of research. With the exception of a single, albeit important, issue regarding the "Informed Consent Statement" (see below), all other concerns have been fully and impressively addressed:

In the response letter, the authors state, "We've removed the term 'informed consent' to avoid implying a formal research consent process."  However, the revised manuscript includes a dedicated section on page 17 titled "Informed Consent Statement" which begins, "Informed consent was obtained from all users...".  This directly contradicts the authors' stated revision, the IRB's determination that the study was not human subjects research, and the more accurate description of opt-in procedures provided in the methods section.  Such a statement is misleading and must be revised.  It is strongly recommended that the "Informed Consent Statement" be removed or retitled (e.g., "User Consent and Data Privacy") and rephrased to accurately reflect Viamo's standard user agreements and the opt-in process for voluntary data, avoiding the formal term "informed consent."

Once this change is made, the manuscript will represent a valuable contribution to the literature on digital mental health in low-resource settings.

Author Response

Dear Reviewer,

Thank you again for all of your thoughtful feedback.

We overlooked the "Informed Consent Statement" on page 17 and acknowledge that it was misleading given the IRB determination. We have now replaced the section with a revised heading, User Consent and Data Privacy, and updated the language to accurately reflect Viamo’s opt-in process and standard data privacy protocols, without implying a formal research consent process. We hope this addresses your concern appropriately.

Warm regards,
Chris Barkley